# Hepatitis B, C, and D Virus Infection among Population Aged 10–64 Years in Mongolia: Baseline Survey Data of a Nationwide Cancer Cohort Study

**DOI:** 10.3390/vaccines10111928

**Published:** 2022-11-14

**Authors:** Davaalkham Dambadarjaa, Otgonbayar Radnaa, Ser-Od Khuyag, Oyu-Erdene Shagdarsuren, Uranbaigali Enkhbayar, Yerkyebulan Mukhtar, Enkh-Oyun Tsogzolbaatar, Gunchmaa Nyam, Shatar Shaarii, Pramil Singh, Masaharu Takahashi, Bira Namdag, Hiroaki Okamoto

**Affiliations:** 1School of Public Health, Mongolian National University of Medical Sciences, Ulaanbaatar 14210, Mongolia; 2School of Medicine, Mongolian National University of Medical Sciences, Ulaanbaatar 14210, Mongolia; 3Transdisciplinary Tobacco Research Program, Loma Linda University Cancer Center, Loma Linda, CA 92354, USA; 4Division of Virology, Department of Infection and Immunity, Jichi Medical University School of Medicine, Shimotsuke 329-0498, Japan

**Keywords:** hepatitis B, hepatitis C, infection, virus, Mongolia, prevalence

## Abstract

Hepatitis B, C, and D virus infections are a major public health problem, and Mongolia has one of the highest prevalences of dual and triple infections in the world. We aimed to determine the seroprevalence of hepatitis infection and dual or triple hepatitis infections among 10–64-year-olds. A questionnaire was used to identify risk factors for hepatitis infection, and seromarkers were measured by the fully automated immunologic analyzer HISCL-5000. Among a total of 10,040 participants, 8.1% of the population aged 10–64 was infected with HBV, 9.4% with HCV, and 0.4% with HBV and HCV, and the prevalence of the disease varied by age, sex, and the area of residence. Young people were particularly unaware of their hepatitis infection status. A small proportion of children aged 10 to 19 years and the majority of adults younger than 30 years were unaware of their HBV and HCV infection. Men were also more likely to be unaware of their HBV and HCV infection status than women. The results suggested that the prevalence of infection in the general population is high and that most people are unaware that they are infected or have become chronic carriers. Identifying mono-, co-, or triple-infection status is critical to prevent the rapid progression of liver disease among the Mongolian population.

## 1. Introduction

Hepatitis B virus (HBV) and hepatitis C virus (HCV) are the leading causes of liver cirrhosis and hepatocellular carcinoma (HCC) and are major public health problems worldwide [1]. The number of people living with chronic HBV infection in 2019 was estimated at 296 million worldwide, and 1.5 million people become newly infected each year [2]. Chronic HCV infection was prevalent in 58 million people and resulted in approximately 290,000 deaths [3]. In 2016, the World Health Organization (WHO) developed a strategy to eliminate viral hepatitis and aimed to reduce new infections by 90% and deaths caused by hepatitis infection by 65% by 2030, in line with the Sustainable Development Goals [4]. Key priorities included reducing the transmission of HBV and HCV infections by scaling up screening, organizing resources, and developing strategies to support countries.

Hepatitis virus infection is considered endemic in Mongolia, which has one of the highest rates of HCC deaths per 100,000 population, eight times higher than the global average [5]. Previous studies have found the prevalence of HBV surface antigen (HBsAg) in 9.3% of the relatively healthy population, 5.2–9.8% of children, and 7.8% of blood donors [6,7,8]. Moreover, antibodies to HCV (anti-HCV) were detected in 11–16% of the general population, which is significantly higher compared to other countries [9,10].

In addition, it has been reported that up to 4.5% of HBV patients worldwide are also infected with hepatitis D virus (HDV) [11], and Mongolia had the highest prevalence of HDV infection at 36.9% in 2020 [12]. HDV is a virus that requires HBV for replication, and their coinfection is considered the most severe form of viral hepatitis, leading to the necrosis of liver tissue and a higher risk of HCC [13]. Hemodialysis patients, people who inject drugs, and people with HCV or HIV infection are more likely to contract HDV infection, and it is known that HDV leads to HCC in 20% of cases and cirrhosis in 18% of cases [12].

In a recent study using models for 110 countries, the prevalence of viremic HCV infection worldwide was 0.7%, with 56.8 million infections [14]. Mongolia had the highest HCV prevalence globally in both 2015 (6.4%) and 2020 (4.2%), followed by Pakistan and Ukraine. HCV is mainly transmitted via blood; thus, injection drug users, people who receive tattoos or manicure and pedicure services, and hemodialysis patients are at higher risk. In addition, HCV is transmitted vertically from mother to child and through sexual intercourse [15]. HCV is known to be the major cause of cirrhosis, HCC, and liver transplantation. In approximately 20% of HCV infections, the disease progresses to cirrhosis and end-stage liver disease within 10–20 years.

Chronic viral hepatitis infections are responsible for an estimated 57% of liver cirrhosis and 78% of primary liver cancer cases [16]. Studies have shown that 93% of liver cancer patients in Mongolia are infected with hepatitis, and that one in ten deaths is due to liver cancer, accounting for 38.5% of all cancers in Mongolia and 43.8% of all cancer deaths [17]. Most cases are diagnosed in the late stages of cancer due to the asymptomatic nature of hepatitis infection, and a large proportion of patients die within a year of diagnosis. Most treatments for liver cancer in Mongolia are currently palliative.

Effective and safe hepatitis B vaccines have been implemented since 1982 and have contributed significantly to the prevention of both infection and chronic liver disease [2]. Moreover, direct antiviral agents (DAAs) to eliminate HCV infection and the antiviral treatment of HBV have greatly reduced the transmission and incidence of hepatitis and its associated complications. Furthermore, measures to reduce contact with infected blood, blood products, and blood-contaminated items can curb the spread of HBV and HCV [18]. However, most people in Mongolia do not have access to treatment due to a lack of screening and clinical services and high prices for antiviral drugs. Although the cost of drugs to treat hepatitis B and C has been reduced by nearly 90%, these drugs are still unaffordable for most Mongolians. In addition, the COVID-19 pandemic has further impeded access to treatment for hepatitis infection and delayed progress toward achieving the program goals of WHO.

Determining the prevalence of hepatitis infection in the general population is critical for planning appropriate clinical care and effective public health interventions, formulating evidence-based policies, and introducing new treatment options. Our objective was to investigate the prevalence of HBV, HCV, and HDV infections in a relatively healthy Mongolian population as a baseline survey in a multiyear cohort study to support the Sustainable Development Goals to reduce hepatitis infections.

## 2. Materials and Methods

### 2.1. Study Design and Population

This study was conducted as part of a baseline study of the National Cancer Center Cohort Survey. Mongolia is geographically divided into the western, Khangai, eastern, and central regions. Two provinces from each region and the capital city, Ulaanbaatar, were selected. The appropriate sample size was estimated at 9800 people aged 10 to 64 years, based on 2016 demographic data from the National Statistics Office. To obtain a socioeconomically and geographically representative sample, a multistage random-sampling method was used, and the population was selected from the following areas: the Bayanzurkh, Songinokhairkhan, and Chingeltei districts in Ulaanbaatar; the Gobi-Altai and Uvs provinces in the western region; the Arkhangai and Khuvsgul provinces in the Khangai region; Dornogobi and Umnugobi in the Tuv region; and Sukhbaatar in the eastern region. A list of the names of residents aged 10 to 64 years was compiled from the registers of each province and district to randomly select participants. Inclusion criteria included individuals aged 10 to 64 years who held Mongolian citizenship and agreed to participate in the study. Individuals who did not meet these criteria were excluded from the study. Participants were then invited by telephone to their local health centers, where they were interviewed with a questionnaire and blood samples were collected.

### 2.2. Data Collection and Laboratory Analyses

The study was conducted between 2017 and 2018 by questionnaire completion and laboratory analysis. The questionnaire contained closed and semi-closed questions from 12 categories and 188 questions and was completed in a face-to-face interview. Using a pre-prepared questionnaire, the researchers asked the respondents to circle the codes for the relevant answers.

Blood samples were collected from consenting participants, stored at −20°C, and transported for further analysis. A fully automated HISCL-5000 chemiluminescence enzyme immunoassay (CLEIA) immunological analyzer (Sysmex, Japan) was used to measure HBsAg, HBV-e antigen (HBeAg), HBV-e antibody (anti-HBe), antibodies against HBV core antigen (anti-HBc), and anti-HCV according to the manufacturer’s protocol. The sensitivity of this method is 100%, and the specificity is 99.94%. HBV infection was determined by quantitative HBsAg (qHBsAg). This marker reflects the cccDNA concentration in hepatocytes and is useful for the correct determination of disease status and the risk of cancer development, as well as for monitoring the efficacy of antiviral therapy. Anti-HDV immunoglobulin G (IgG) was analyzed by in-house enzyme-linked immunosorbent assay (ELISA) using purified recombinant S-HDAg protein expressed in silk worm pupae [19] in the laboratory of the Division of Virology, Department of Infection and Immunity, Jichi Medical University, Japan. The specificity of the anti-HDV ELISA was confirmed by an absorption assay in accordance with previously described methods [19]. Only participants who tested positive for HBsAg were further assessed for HDV, as this only occurs in people infected with HBV.

### 2.3. Statistical Analysis

Quantitative variables were assessed for normality by the Kolmogorov–Smirnov test. Categorical variables, expressed as percentages, were compared using the chi-square test and Fisher’s exact test when appropriate. All statistical analyses were performed on SPSS 25.0 (IBM corp., Armonk, NY, USA).

### 2.4. Ethical Considerations

Ethical Review Committee approval (protocol number 22/1A) was obtained from the Mongolian National University of Medical Sciences for the study protocol, and written informed consent forms were filled out by all study participants before blood collection and questionnaire administration. For children aged under 16 years, informed consent forms were signed by a parent or a legal guardian. The study conformed to the regulations of the Declaration of Helsinki.

## 3. Results

A total of 10,040 people aged 10 to 64 years from four regions and eight provinces participated in the survey. The sample was representative of the general population. Of the total participants, 4008 (39.9%) were male, 6601 (65.7%) lived in the urban areas, and 3430 (34.3%) were from the rural provinces.

Approximately 80% of our study participants tested negative for hepatitis infection, while the seroprevalence of HCV (8.9%) infection was the highest among those who tested positive (Table 1). Participants who were positive for HBV infection were from urban areas (4.9%), the western region (5.7%), male (5.6%), and aged between 30 and 49 years (58.1%). As for HCV infection, most were of rural residence (10.2%), from the Khangai region (11.3%), female (10.4%), and were aged between 30 and 49 years (47.7%). HDV infection distribution was similar to HCV infection except for the higher seroprevalence among males (2.7%) than females (2.1%). When the prevalence of HBV, HCV, or HDV coinfection in the Mongolian population aged 10–64 years was examined, 0.3% (*n* = 38) had a coinfection of HBV and HCV, and 4.8% had an HBV and HDV coinfection. A triple infection of HBV, HDV, and HCV seroprevalence was observed in 10 participants (0.2%). Overall, co- and triple infections were higher among those who resided in rural area and were older. HBV and HCV coinfection was more likely to be prevalent among males, while triple infections were prevalent among females. The western and Khangai regions had the highest seroprevalence of hepatitis infections.

Among the participants, 275 tested positive for both HBsAg and anti-HDV (Table 2). Those who had an HBV and HDV coinfection were significantly different to those who only had an HBV infection in terms of residence (*p* = 0.001), region (*p* = 0.001), age group (<0.0001), and hepatitis infection status (*p* < 0.0001). To elaborate, those of rural residence, living in the eastern region, aged between 30 and 39 years, and testing positive for HBV infection were more likely to test positive for anti-HDV. There was no significant difference observed between genders.

Laboratory analysis showed that HBeAg was positive in 14.7%, anti-HBe in 78.6%, anti-HBc in 43.8%, and anti-HBs in 41.1% of participants (Table 3). This indicated that 43.8% of the population aged 10–64 years were already infected with HBV, 41.1% were immune to infection, and 14.7% of chronic HBV carriers had an active infection. Hepatitis marker results varied by age, sex, and the area of residence. The proportion of individuals with HBeAg-positive serology results was higher in men (*p* = 0.031), while the proportion of individuals with positive markers of anti-HBc (*p* = 0.001) and anti-HBs (*p* = 0.001) was higher in rural areas.

When participants’ knowledge of their hepatitis infection diagnoses was examined, 64.8% of individuals with HBV and 62.7% of individuals with HCV did not know they were infected (Table 4). Young people were particularly unaware of their hepatitis infection status. A small proportion of children aged 10 to 19 years and the majority of adults younger than 30 years were unaware of their HBV and HCV infection. Men were also more likely to be unaware of their HBV and HCV infection status than women. The findings suggest that the prevalence of infection in the general population is high and that most are unaware that they are infected or have become chronic carriers.

## 4. Discussion

In our study, we examined more than 10,000 individuals aged 10 to 64 years from eight provincial centers and the capital city of Ulaanbaatar to determine the seroprevalence of HBV, HCV, and HDV infection. A multistage sampling method was used to obtain a sample that was representative of all age groups, sociodemographic characteristics, and geographic regions. The prevalence of HBV and HCV infection and coinfection was high in this population. A meta-analysis of 27 studies conducted in China in 2013–2017 found that 6.89% (95% CI: 5.84–7.95%) of the relatively healthy population were infected with HBV, and it was estimated that 84 million people in the total population were infected with HBV in 2018 [20]. The prevalence of HBV infection in China was significantly higher in rural areas (5.9%, *p* < 0.001) and in men (5.9%) than in women (5.0%) and residents of urban areas (3.3%), which was similar to the results of our study. In addition, the prevalence of HBV infection was relatively high in the population older than 20 years compared with children, consistent with our findings. A systematic review of 1800 studies published between 1965 and 2013 that examined HBV infection in 161 countries found that the prevalence of HBV infection worldwide was 3.61% [21]. In the Western Pacific region, the prevalence of HBV was 5.26%, and in Africa, the prevalence was 8.83%. However, the prevalence of HBV and HCV infection in relatively healthy people in 13 European countries was 0.1–4.4% and 0.1–5.9%, respectively, lower than in our study [22,23].

In a study investigating the prevalence of HBV and HDV infections in 249 Mongolians, 9.6% were HBsAg-positive, of whom 13.5% were men and 5.7% were women [24]. The comparatively lower prevalence of HBV infection can be explained by the different study periods and age groups of the studied populations. According to a meta-analysis of 54 international studies conducted between 1973 and 2016, the prevalence of HCV infection in relatively healthy people was 0.68%, which was much lower than in our study [25]. The highest prevalence of HCV infection in Asia was in Uzbekistan (11.3%), and the lowest was in South Korea (0.8%). In our study, the prevalence of coinfection with HBV and HCV was 0.4%. However, in a 2000–2016 study in India, the prevalence of HBV and HCV coinfection was 1.89%, which was much higher than in our study (0.6%) [26].

In a 2016–2017 survey of 1079 people in South Asian countries, including Myanmar, Thailand, Vietnam, Cambodia, the Philippines, and Singapore, 46.0% did not know they had an HBV infection [27]. Similarly to our study, 69% of HBV-infected persons and 53.5% of HCV-infected persons in the United States did not know they were infected [28]. The lack of awareness around participants’ diagnoses underscores the importance of health education, early detection, screening, and intervention among young adults.

Cancer morbidity and mortality have steadily increased in Mongolia in recent decades, with liver cancer morbidity and mortality predominating in both men and women. Because HBV and HCV infections are risk factors for liver cirrhosis and HCC, it is imperative that appropriate public health measures be taken to control these viral infections. HBV infection can be prevented by vaccination, but there is no specific prevention for HCV infection other than avoiding contact with contaminated blood.

Researchers in Mongolia have continuously provided important evidence and information about HBV, HCV, and HDV infections [29,30,31]. The strength of our study is that we collected recent data (2017–2018) and included a large sample of 10,000 individuals that was representative of the nationwide population aged 10–64 years; additionally, the sample included residents of 8 provinces (a total of 21 provinces) and 6 districts of the capital city (a total of 9 districts), thus reflecting the geographic regional characteristics and age distribution. Moreover, hepatitis virus testing and diagnosis are performed in Mongolia using various diagnostic methods. Immunochromatographic analysis (ICA rapid test) is widely used, and enzyme-linked immunosorbent assay (ELISA) and chemiluminescent enzyme immunoassay (CLEIA) are used for repeat testing or to confirm diagnosis. In our study, we detected HBV and HCV infections with the highly sensitive CLEIA, which was a novel approach novel and represented a strength of our study. The serological markers of anti-HCV, qHBsAg, anti-HBs, HBeAg, anti-HBe, and anti-HBc IgG were measured using the fully automated HISCL-5000 (Sysmex, Japan) CLEIA system to detect and confirm HBV and HCV infections.

Viral hepatitis infections can be prevented by administering vaccines, improving hospital infection prevention and control procedures, and educating the public. Mongolia was one of the first countries to include hepatitis B vaccination in routine immunization schedules for newborns and children under one year of age in 1991. Despite Mongolia’s large nomadic population, an estimated 99% of newborns receive the first dose of the vaccine within 24 h of birth. The incidence of viral hepatitis B has declined significantly since this vaccination was introduced. However, our findings suggest that the number of hepatitis infections in Mongolia remains high and that vaccination coverage needs to be increased to reduce HBV infections. Therefore, measures such as vaccinating newborns within 24 h of birth, the timely administration of other vaccines, and maintaining vaccine quality during transport and storage are critical. Other measures to reduce and prevent the spread of HBV and HCV remain important in all settings. These include protecting the healthy population from risk factors for infection, health education and spreading awareness, the screening and early detection of infection, and the prevention of complications.

## Figures and Tables

**Table 1 vaccines-10-01928-t001:** Distribution of hepatitis serological antigens in the study population.

	Negative	HBV Mono-Infection (HBsAg)	HCV Mono-Infection (Anti-HCV Positive)	HBV and HCV	HBV and HDV (Anti-HDV Positive)	HBV, HDV, and HCV	*p*-Value
	N (%) Row	N (%) Row	N (%) Row	N (%) Row	N (%) Row	N (%) Row	
Total	4115 (78.8%)	227 (4.3%)	463 (8.9%)	17 (0.3%)	250 (4.8%)	10 (0.2%)	
Residence							<0.0001
Rural	1969 (76.9%)	96 (3.8%)	262 (10.2%)	9 (0.4%)	143 (5.6%)	8 (0.3%)	
Urban	2146 (80.7%)	131 (4.9%)	201 (7.6%)	8 (0.3%)	107 (4.0%)	2 (0.1%)	
Region							<0.0001
Western	228 (72.2%)	18 (5.7%)	31 (9.8%)	2 (0.6%)	23 (7.3%)	2 (0.6%)	
Khangai	722 (78.6%)	31 (3.4%)	104 (11.3%)	4 (0.4%)	33 (3.6%)	2 (0.2%)	
Central	739 (79.0%)	31 (3.3%)	92 (9.8%)	3 (0.3%)	43 (4.6%)	3 (0.3%)	
Eastern	280 (72.0%)	16 (4.1%)	35 (9.0%)	0 (0.0%)	44 (11.3%)	1 (0.3%)	
Ulaanbaatar	2146 (80.7%)	131 (4.9%)	201 (7.6%)	8 (0.3%)	107 (4.0%)	2 (0.1%)	
Gender							<0.0001
Male	1728 (78.0%)	125 (5.6%)	150 (6.8%)	5 (0.2%)	138 (6.2%)	1 (0.0%)	
Female	2387 (79.4%)	102 (3.4%)	313 (10.4%)	12 (0.4%)	112 (3.7%)	9 (0.3%)	
Age group							<0.0001
10–29	1791 (43.5%)	74 (32.5%)	43 (9.3%)	2 (11.7%)	38 (15.2%)	1 (10.0%)	
30–49	1804 (43.8%)	132 (58.1%)	221 (47.7%)	10 (5.8%)	155 (62.0%)	2 (20.0%)	
50–64	520 (12.6%)	21 (9.2%)	199 (42.9%)	5 (29.4%)	57 (22.8%)	7 (70.0%)	

**Table 2 vaccines-10-01928-t002:** Prevalence of HDV infections among population aged 10–64 years in Mongolia (n = 5312).

	HBV and HDV Coinfection		*p*-Value
	Negative n (%) Row	Positive n (%) Row	
Total	5036 (94.8%)	276 (5.2%)	
HBV (positive)	240 (47.4%)	266 (52.6%)	
Residence			0.007
Rural	2338 (93.9%)	151 (6.1%)	
Urban	2698 (95.6%)	125 (4.4%)	
Gender			0.0001
Male	2097 (93.4%)	148 (6.6%)	
Female	2939 (95.8%)	128 (4.2%)	
Region			0.0001
Western	280 (91.8%)	25 (8.2%)	
Khangai	861 (96.1%)	35 (3.9%)	
Central	865 (95.0%)	46 (5.0%)	
Eastern	332 (88.1%)	45 (11.9%)	
Ulaanbaatar	2698 (95.6%)	125 (4.4%)	
Age group			0.0001
10–29	1938 (98.0%)	39 (2.0%)	
30–49	2289 (93.2%)	166 (6.8%)	
50–64	809 (91.9%)	71 (8.1%)	

**Table 3 vaccines-10-01928-t003:** Identification of HBV markers among study participants by demographic characteristics.

Variable	HBeAg (+) ^§^	Anti-Hbe ^§^ (+)	Anti-HBc ^†^ (+)	Anti-HBs ^†^ (+)
	N (%)	N (%)	N (%)	N (%)
Gender	*p* = 0.031	*p* = 0.674	*p* = 0.807	*p* = 0.0001
Male	61 (17.4)	272 (77.9)	1732 (43.6)	1414 (37.5)
Female	33 (11.4)	230 (79.3)	2631 (43.9)	2519 (43.4)
Age group	*p* = 0.557	*p* = 0.0001	*p* = 0.0001	*p* = 0.0001
10–19	6 (18.2)	7 (21.2)	87 (4.0)	665 (31.0)
20–29	23 (19.7)	87 (74.4)	632 (34.3)	624 (35.1)
30–39	29 (12.9)	187 (83.9)	1570 (60.2)	1124 (45.5)
40–49	20 (12.9)	130 (83.9)	1092 (60.6)	822 (48.1)
50–59	14 (15.4)	75 (82.4)	787 (62.1)	571 (48.0)
60<	2 (10.0)	16 (80.0)	195 (66.1)	127 (45.7)
Residence	*p* = 0.127	*p* = 0.443	*p* = 0.001	*p* = 0.0001
Rural	44 (12.7)	275 (79.7)	1585 (46.2)	1523 (44.4)
Urban	50 (17.0)	227 (77.2)	2778 (42.5)	2410 (39.2)
Total	94 (14.7)	502 (78.6)	4363 (43.8)	3933 (41.1)

^§^ HBeAg and anti-HBe markers were tested among 939 HBsAg-positive subjects. ^†^ Anti-HBc and anti-HBs markers were tested in all study participants (n = 10,040).

**Table 4 vaccines-10-01928-t004:** Awareness of HBV, HCV, and HDV infection status.

Variable	Knows Infection Status of HBV		Knows Infection Status of HCV		Knows Infection Status of HDV	
	No N (%) Row	Yes N (%) Row	No N (%) Row	Yes N (%) Row	No N (%) Row	Yes N (%) Row
Gender						
Male	328 (67.2%)	160 (32.8%)	178 (68.7%)	81 (31.3%)	147 (99.3%)	1 (0.7%)
Female	264 (58.5%)	187 (41.5%)	349 (60.1%)	232 (39.9%)	126 (98.4%)	2 (1.6%)
Age group						
10–14	19 (100.0%)	0 (0.0%)	7 (100.0%)	0 (0.0%)	1 (100.0%)	0 (0.0%)
15–19	13 (100.0%)	0 (0.0%)	7 (100.0%)	0 (0.0%)	0 (0.0%)	0 (0.0%)
20–24	28 (80.0%)	7 (20.0%)	11 (100.0%)	0 (0.0%)	5 (100.0%)	0 (0.0%)
25–29	69 (53.5%)	60 (46.5%)	35 (74.5%)	12 (25.5%)	32 (97.0%)	1 (3.0%)
30–34	113 (61.4%)	71 (38.6%)	59 (68.6%)	27 (31.4%)	52 (98.1%)	1 (1.9%)
35–39	91 (61.1%)	58 (38.9%)	65 (61.9%)	40 (38.1%)	39 (97.5%)	1 (2.5%)
40–44	73 (58.4%)	52 (41.6%)	67 (58.8%)	47 (41.2%)	33 (100.0%)	0 (0.0%)
45–49	69 (67.0%)	34 (33.0%)	74 (69.2%)	33 (30.8%)	40 (100.0%)	0 (0.0%)
50–54	61 (63.5%)	35 (36.5%)	94 (62.3%)	57 (37.7%)	40 (100.0%)	0 (0.0%)
55–59	35 (63.6%)	20 (36.4%)	70 (56.0%)	55 (44.0%)	22 (100.0%)	0 (0.0%)
60–64	21 (67.7%)	10 (32.3%)	38 (47.5%)	42 (52.5%)	9 (100.0%)	0 (0.0%)
Total	592 (63.0%)	347 (37.0%)	527 (62.7%)	313 (37.3%)	273 (98.9%)	3 (1.1%)

## Data Availability

Data are available upon reasonable request.

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
