# Peer review of "Hepatitis B, C, and D Virus Infection among Population Aged 10–64 Years in Mongolia: Baseline Survey Data of a Nationwide Cancer Cohort Study"

_vaccines, 2022, doi:10.3390/vaccines10111928_

Round 1
Reviewer 1 Report
I have reviewed the manuscript entitled “Hepatitis B, C and D virus infection among the population aged 10-64 years in Mongolia: Baseline survey data of a nationwide cancer cohort study” with great interest. It is an important manuscript with public health importance specifically in the ear of hepatitis elimination. However, the manuscript may need significant modification to ensure clarity of information and methods used for the study.
Materials and methods:
1. The author should describe the participant selection and enrollment process in detail, e.g., how the participant was selected, identified, and enrolled, enrollment site, consent process, etc.
2. The 188 question document should be submitted as a supplementary file.
3. Based on which test, a patient was considered HBV positive? Generally, the US CDC has a recommendation and the authors should follow it.
4. It is not clear if all tests were done in all samples e.g., HBV tests and HDV tests
5. It is not clear if the recombinant S- 118 HDAg protein ELISA is a validated test, there is no reference to previous publication on this protein.
Results:
1. How many participants were approached and what was the refusal rate?
2. Table 1 is confusing, plz clarify that HBV, HCV, and HDV in columns 2, 3, and 4 are mono-infections or what? 78.8% negative of 10040 participants should be 7911, not 4115.
3. The author should define how many % of HBV patients had HDV infection, the calculations do not add up to the total.
4. It is not clear if table 2 is comparing HBV positive and HBV+HDV positive patients or something else
Discussion:
1. Discussion should be more concise and focused on the result section and its interpretations.
Author Response
Response to Reviewer 1 Comments
I have reviewed the manuscript entitled “Hepatitis B, C and D virus infection among the population aged 10-64 years in Mongolia: Baseline survey data of a nationwide cancer cohort study” with great interest. It is an important manuscript with public health importance specifically in the ear of hepatitis elimination. However, the manuscript may need significant modification to ensure clarity of information and methods used for the study.
Materials and methods
Point 1: The author should describe the participant selection and enrollment process in detail, e.g., how the participant was selected, identified, and enrolled, enrollment site, consent process, etc.
Response 1: We accordingly included the participant selection and enrolment process in the page 3, line number 105: “A list of resident names aged 10 to 64 years was obtained from each province and district registries to randomly select participants. Inclusion criteria included those aged 10 to 64 years, Mongolian citizen, and consented to participate. Individuals who did not meet these criteria were excluded from the study. The participants were then invited through a phone call to their local health centers for questionnaire interviews and blood sample collection.”
Point 2: The 188 questions document should be submitted as a supplementary file.
Response 2: The document has been uploaded as a supplementary file.
Point 3: Based on which test, a patient was considered HBV positive? Generally, the US CDC has a recommendation and the authors should follow it.
Response 3: We followed the instructions of the World Health Organization’s “Guideline on Hepatitis B and C testing.” For reference, please visit this website:
http://apps.who.int/iris/bitstream/handle/10665/251330/WHO-HIV-2016.23-eng.pdf;sequence=1
Point 4: It is not clear if all tests were done in all samples e.g., HBV tests and HDV tests
Response 4: We apologize for the confusion. HDV tests were performed only in study participants who tested positive for HBV as HDV infection can only occur in those infected with HBV. A sentence was added to the “2.2. Data collection and laboratory analyses” section.
Point 5: It is not clear if the recombinant S-118 HDAg protein ELISA is a validated test, there is no reference to previous publication on this protein.
Response 5: The relevant reference (Inoue J, et al. J Med Virol. 2005;76(3):333-40. doi: 10.1002/jmv.20363. PMID:15902700) was cited in the text. The validity of the assay has been verified in the HDV-viremic samples.
Results
Point 1: How many participants were approached and what was the refusal rate?
Response 1: Approximately 15000 individuals were invited to participate in the study and the refusal rate was 33%.
Point 2: Table 1 is confusing, plz clarify that HBV, HCV, and HDV in columns 2, 3, and 4 are mono-infections or what? 78.8% negative of 10040 participants should be 7911, not 4115.
Response 2: We are sorry for the confusion. Columns 2, 3, and 4 indicate mono-infections, the clarification was added to Table 1 accordingly. Please note that the percentages are shown in rows and not columns, so 4115 would be correct.
Point 3: The author should define how many % of HBV patients had HDV infection, the calculations do not add up to the total.
Response 3: Overall, 227 participants had HBV mono-infection while 250 participants had both HDV and HBV infection. Among the participants with HBV infection, 52,4% had HDV infection.
Point 4: It is not clear if table 2 is comparing HBV positive and HBV+HDV positive patients or something else
Response 4: We apologize for the confusion. Table 2 is comparing HBV+HDV positive patients to other variables. We changed the subheading to ‘HBV and HDV co-infection’ accordingly.
Discussion
Point 1: Discussion should be more concise and focused on the result section and its interpretations.
Response 1: We thank you for the suggestion. The discussion section has been edited as recommended.

Reviewer 2 Report
1.the validity and reliability of this study should have a description in the content.
2.the data collectin process of questionnaire should have a detail description, and how conduct the questionnaire interview.
3. the representative of study samples should have a discussion.
4. Mnay samples are too young to understand the "awareness of HBV and HCV" approriately. Maybe the young age group should exclude in the study population.1.the validity and reliability of this study should have a description in the content.
Author Response
Response to Reviewer 2 Comments
Thank you so much for very valuable comments given by the reviewers and a chance for us to revise the manuscript, entitled ‘Hepatitis B, C and D virus infection among the population aged 10-64 years in Mongolia: Baseline survey data of a nationwide cancer cohort study’, according to the reviewer’s comments. The revised parts are written in bold
- The validity and reliability of this study should have a description in the content
Response: We have added validity and reliability of the study in the method section in page 3, line 121-124.
- The data collection process of questionnaire should have a detail description, and how conduct the questionnaire interview
Response: we have added a data collection process of questionnaire in method section in page3, line 115-117.
- The representative of study samples should have a discussion
Response: We have added the representative of the study samples in discussion section in page 7, line 242-244.
- Many samples are too young to understand “awareness of HBV and HCV” appropriately. Maybe the young age group should exclude in the study population.
Response: We have excluded the young age group (10-14) in result section